# MGMT genomic rearrangements contribute to chemotherapy resistance in gliomas

Barbara Oldrini[1,9], Nuria Vaquero-Siguero[1,9], Quanhua Mu[2,9], Paula Kroon[1], Ying Zhang[3], Marcos Galán-Ganga[1], Zhaoshi Bao [2,3], Zheng Wang[3], Hanjie Liu[3], Jason K. Sa [4,8], Junfei Zhao[5], Hoon Kim [6], Sandra Rodriguez-Perales[7], Do-Hyun Nam[4], Roel G. W. Verhaak [6], Raul Rabadan [5], Tao Jiang [3,10✉], Jiguang Wang [2,10✉] & Massimo Squatrito [1,10✉]

Temozolomide (TMZ) is an oral alkylating agent used for the treatment of glioblastoma and is now becoming a chemotherapeutic option in patients diagnosed with high-risk low-grade gliomas. The O-6-methylguanine-DNA methyltransferase (MGMT) is responsible for the direct repair of the main TMZ-induced toxic DNA adduct, the O6-Methylguanine lesion. *MGMT* promoter hypermethylation is currently the only known biomarker for TMZ response in glioblastoma patients. Here we show that a subset of recurrent gliomas carries *MGMT* genomic rearrangements that lead to MGMT overexpression, independently from changes in its promoter methylation. By leveraging the CRISPR/Cas9 technology we generated some of these *MGMT* rearrangements in glioma cells and demonstrated that the *MGMT* genomic rearrangements contribute to TMZ resistance both in vitro and in vivo. Lastly, we showed that such fusions can be detected in tumor-derived exosomes and could potentially represent an early detection marker of tumor recurrence in a subset of patients treated with TMZ.

[1] Seve Ballesteros Foundation Brain Tumor Group, Molecular Oncology Programme, Spanish National Cancer Research Center, CNIO, 28029 Madrid, Spain. [2] Division of Life Science, Department of Chemical and Biological Engineering, Center of Systems Biology and Human Health and State Key Laboratory of Molecular Neuroscience, The Hong Kong University of Science and Technology, Hong Kong, China. [3] Beijing Neurosurgical Institute, Capital Medical University, 100050 Beijing, China. [4] Department of Neurosurgery, Samsung Medical Center, Sungkyunkwan University School of Medicine, Seoul 06531, Korea. [5] Department of Systems Biology, Columbia University, New York, NY 10032, USA. [6] The Jackson Laboratory for Genomic Medicine, Farmington, CT 06032, USA. [7] Molecular Cytogenetics Group, Human Cancer Genetics Program, Spanish National Cancer Research Center, CNIO, 28029 Madrid, Spain. [8] Present address: Department of Biomedical Sciences, Korea University College of Medicine, Seoul, Korea. [9] These authors contributed equally: Barbara Oldrini, Nuria Vaquero-Siguero, Quanhua Mu. [10] These authors jointly supervised this work: Tao Jiang, Jiguang Wang, Massimo Squatrito. ✉email: taojiang1964@163.com; jgwang@ust.hk; msquatrito@cnio.es

The therapeutic benefits of TMZ depend on its ability to methylate DNA, which takes place at the N-7 and O6 positions of guanine and N-3 position of adenine. Although the minor product O6-methylguanine (O6-meG) accounts for <10% of total alkylation, it exerts the greatest potential for apoptosis induction[1]. O6-meG pairs with thymine as opposed to cytosine during DNA replication. The O6-meG:thymine mismatch can be recognized by the post-replication mismatch repair (MMR) system and, according to the futile repair hypothesis, ultimately induces DNA double-strand breaks, cell-cycle arrest, and cell death[2]. The O6-methylguanine-DNA methyltransferase (MGMT) is responsible for the direct repair of O6-meG lesion by transferring the alkyl group from guanine to a cysteine residue. Epigenetic silencing, due to promoter methylation, of the MGMT gene prevents the synthesis of this enzyme, and as a consequence increases the tumors sensitivity to the cytotoxic effects induced by TMZ and other alkylating compounds[3,4]. As today, MGMT promoter hypermethylation is the only known biomarker for TMZ response[4]. However, the discordance between promoter methylation and protein expression detected in a subset of patients limits the prognostic value of methylation assessment[5,6]. Moreover, while MGMT methylation at diagnosis predicts longer survival, this is not the case at recurrence[7]. These evidence would suggest that other mechanisms, in addition to promoter methylation, could contribute to MGMT upregulation in the recurrent tumors[5,7].

According to the 2016 WHO classification, that integrates both histological and molecular features, diffuse gliomas can be divided in IDH-wildtype or IDH-mutant, by the presence of mutations in the isocitrate dehydrogenase 1 and 2 (IDH1/2) genes[8]. TMZ is the standard chemotherapeutic approach in IDH-wildtype gliomas, such as glioblastomas, and, more recently, in high-risk IDH-mutant gliomas[9].

By analyzing a large cohort of IDH-wildtype and mutant recurrent gliomas treated with TMZ, we have discovered that a subset of patients carries distinct MGMT genomic rearrangements. These MGMT alterations lead to MGMT overexpression, independently from changes in its promoter methylation, and contribute to TMZ resistance both in vitro and in vivo.

## Results

**Identification of MGMT gene fusions in recurrent gliomas.** To reveal the landscape of TMZ resistance in glioma patients, we analyzed RNA-sequencing data of 252 TMZ-treated recurrent gliomas, among which 105 (42%) were newly collected (Supplementary Fig. 1a, b and Supplementary Data 1). We then integrated clinical information and performed bioinformatics analysis to determine the mutational status of several key alterations ("Methods").

Overall, we found IDH1 mutation in 38.4% (94 out of 245) patients, 1p/19q co-deletion in 9.4% (23 out of 245) patients, MGMT promoter hypomethylation in 38% (52 out of 136) patients, and DNA hypermutation in 10.7% (27 out of 252) patients (Fig. 1a). By analyzing the RNA-seq data of 252 recurrent gliomas, we identified eight different MGMT fusions in seven patients (~3% of all patients, 95% CI, 1.1–5.6%) (Supplementary Data 1 and Supplementary Table 1). Of note, among the seven patients who harbor MGMT fusions, six are females, which is significantly higher than expected ($P = 0.015$, Fisher exact test, Supplementary Fig. 1c). Importantly, there was significant mutual-exclusiveness between MGMT hypomethylation, DNA hypermutation and MGMT fusion as revealed by a bootstrapping method ($P < 10^{-4}$, see "Methods"), suggesting these alterations were carrying out alternative roles during cancer progression.

Gliomas with MGMT fusions or hypomethylated MGMT promoter had significantly higher MGMT expression, while the DNA hypermutated patients showed the lowest MGMT expression, even lower than the MGMT-methylated tumors (Supplementary Fig. 1d, P-values calculated by Wilcoxon rank-sum test). Interestingly, we found that in IDH-wild-type glioma patients, high MGMT expression indicates worse survival ($P = 0.02$, log-rank test, Supplementary Fig. 1e), while it is associated to a trend of better survival in IDH-mutant patients ($P = 0.04$, log-rank test, Supplementary Fig. 1f). We next performed an in-depth investigation of the eight different MGMT rearrangements: BTRC-MGMT, CAPZB-MGMT, GLRX3-MGMT, NFYC-MGMT, RPH3A-MGMT, and SAR1A-MGMT in HGG, and CTBP2-MGMT and FAM175B-MGMT in LGG (Fig. 1b). Five of the eight partner genes located on chromosome 10q, mostly close to MGMT (Fig. 1b). Interestingly, although the left partners of the MGMT fusions were different, the transcriptomic breakpoint in MGMT invariably located at the boundary of MGMT exon 2, which is 12 bp upstream of the MGMT start codon. In three of the rearrangements (SAR1A-MGMT, RPH3A-MGMT, and CTBP2-MGMT), MGMT coding sequence was fused to the 5′ UTR of the fusion partner. Reconstruction of the chimeric transcripts found all fusions are in-frame, and both the methyltransferase domain and DNA-binding domain of MGMT are intact, suggesting the functions of MGMT might be preserved in the fusion proteins (Fig. 1c). We validated the gene fusions using PCR and Sanger sequencing in samples with enough specimen available (Fig. 1d, e). For one patient (CGGA_1729), we performed whole-genome sequencing (WGS), and analysis of structural rearrangements in this sample revealed a deletion of about 4.8 Mb resulting in the FAM175B-MGMT fusion (Fig. 1f).

**MGMT genomic rearrangements lead to MGMT overexpression.** To characterize the MGMT fusions, we sought to generate some of the identified rearrangements using the CRISPR/Cas9-mediated genome editing. Co-expression of Cas9 with pairs of single-guide RNAs (sgRNAs) has been used to model a variety of chromosomal rearrangements (such as translocations, inversions, and deletions) by creating DNA double-strand breaks at the breakpoints of chromosome rearrangements, which are then joined by non-homologous end joining[10,11]. To generate cell lines carrying the MGMT fusions, we first transduced U251 and U87 cells, two MGMT-methylated GBM cell lines, with lentiviral vectors expressing different combinations of gRNA pairs directed to four different MGMT rearrangements: BTRC-MGMT, NFYC-MGMT, SAR1A-MGMT, and CTBP2-MGMT (Supplementary Fig. 2a–c). The expected chromosomal rearrangements in the bulk populations were detected by PCR at the genomic level and confirmed by Sanger sequencing (Supplementary Fig. 3a, b). The newly generated cell populations were then exposed to TMZ. Surviving clones were observed only in the bulk populations of cells carrying the different fusion events but not in the control cells (sgCtrl, non-targeting sgRNA) (Fig. 2a). We then isolated some of the TMZ-resistance clones and further confirmed the presence of the desired gene fusion by PCR both at the genomic level (Supplementary Fig. 3c) and at mRNA level by reverse transcription PCR (RT-PCR) of cDNA fragments overlapping the fusion exon junctions (Supplementary Figs. 3d–f and 6a). However, we could not confirm at the genomic level the exact breakpoints in the CTBP2-MGMT clones, both in U251 and U87 cells, possibly to the occurrence of larger deletions that removed the binding site of the primers used for our initial studies in the bulk population (Supplementary Fig. 3a, b). Nevertheless, the desired genomic rearrangements were further validated using a break-apart fluorescence in situ hybridization (FISH) assay (Supplementary Fig. 4).

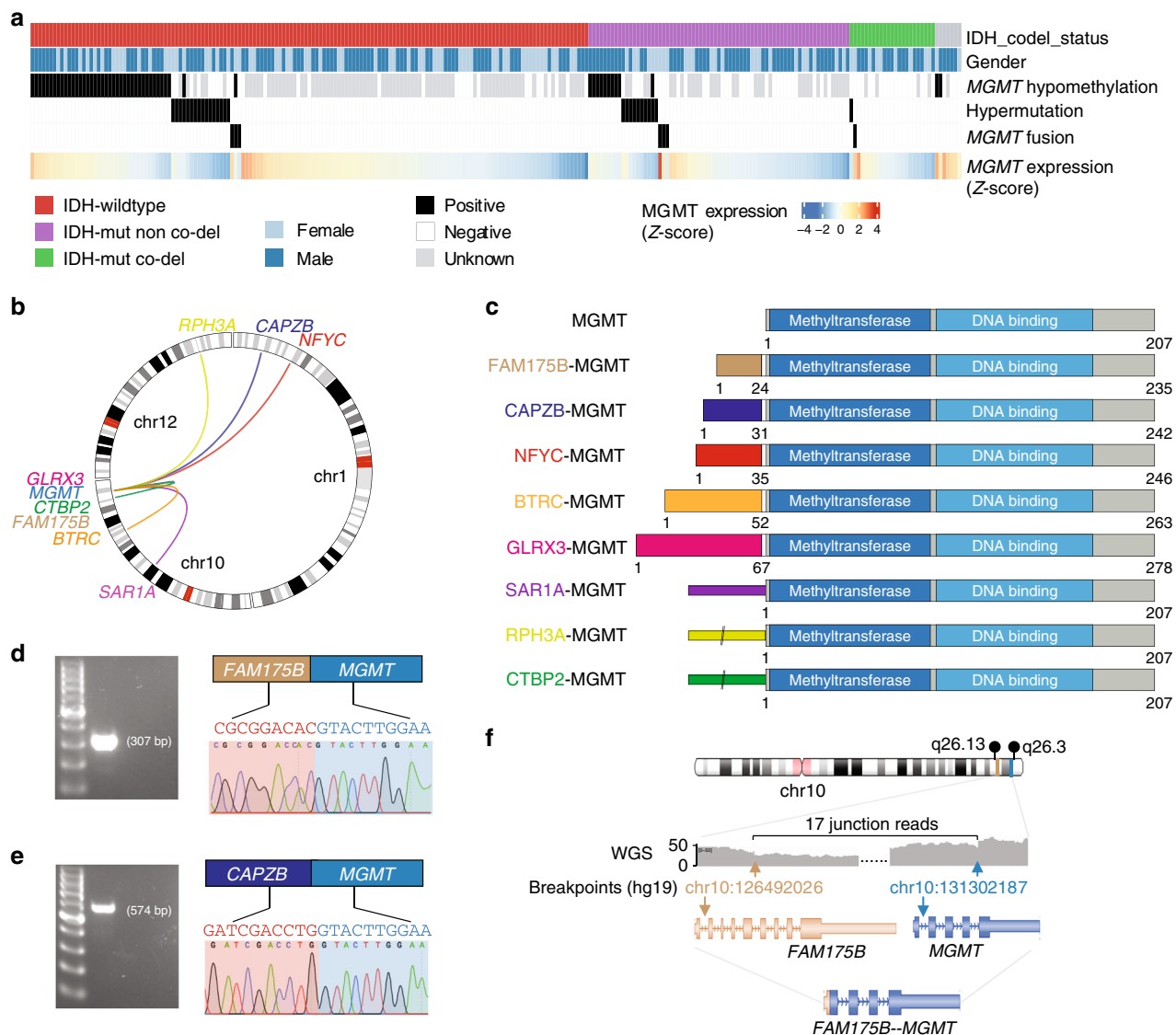

**Fig. 1 Multiple *MGMT* fusions in TMZ-treated recurrent gliomas. a** Landscape of *MGMT* hypomethylation, *MGMT* fusions, DNA hypermutation. **b** Circos plot showing the identified *MGMT* fusions. **c** Structure of the MGMT fusion proteins. Each partner gene is indicated by color, and the narrow bars in SAR1A-MGMT, RPH3A-MGMT, and CTBP2-MGMT mean 5'UTR. **d**, **e** Validation of the *MGMT* fusion genes in positive samples by PCR and Sanger sequencing. The bands in the left panel were validated by Sanger sequencing in the right panel. Limited by specimen availability the validation was performed once. **f** The genomic rearrangement generating *FAM175B-MGMT* fusion. WGS whole-genome sequencing. Source data are provided as a Source Data file.

Promoter exchanges are one class of gene fusions, characterized by the replacement of a gene's regulatory regions with those of another gene, often resulting in deregulation of transcription of the genes participating in the fusion event[12–14]. Another class of gene fusions generates chimeric proteins with biological function different from either of the partner genes from which it originated[12–14]. Since all the MGMT gene fusions identified had similar structures, with the 5′ gene contributing with either small and diverse protein domains or just with the 5′-UTR regions (Fig. 1c), we hypothesized that the TMZ resistance might be driven by increased MGMT expression due to the rearrangements that bring the *MGMT* gene under the control of a more active promoter. Real-time quantitative PCR showed a striking increase of MGMT expression in the clones carrying the different fusions (Fig. 2b), as compared with control cells, without changes in *MGMT* promoter methylation status, as evidenced by methylation-specific PCR (MSP) (Fig. 2c). These results are in

line with what observed in the patient cohort: patients carrying *MGMT* rearrangements showed elevated expression of MGMT, concurrently with a methylated *MGMT* promoter (Fig. 1a; Supplementary Fig. 1d). Western blot analysis, using an anti-MGMT antibody, evidenced a marked overexpression of MGMT at the protein level, especially obvious for the SAR1A-MGMT and CTBP2-MGMT fusion clones (Fig. 2d). Moreover, we observed higher-molecular-weight protein products for BTRC-MGMT and NFYC-MGMT, consistent with the expected size of those fusion proteins. Of note, the different levels of MGMT expression might be determined by the activity of the specific gene's promoter participating in the fusion event and/or by the number of copies of the genomic rearrangement in each specific clone.

To validate our results in an another biologically relevant model of GBM, we also used patient-derived cell line propagated in stem cell medium. These cells, as compared with immortalized cancer cell lines, have been shown to maintain the molecular

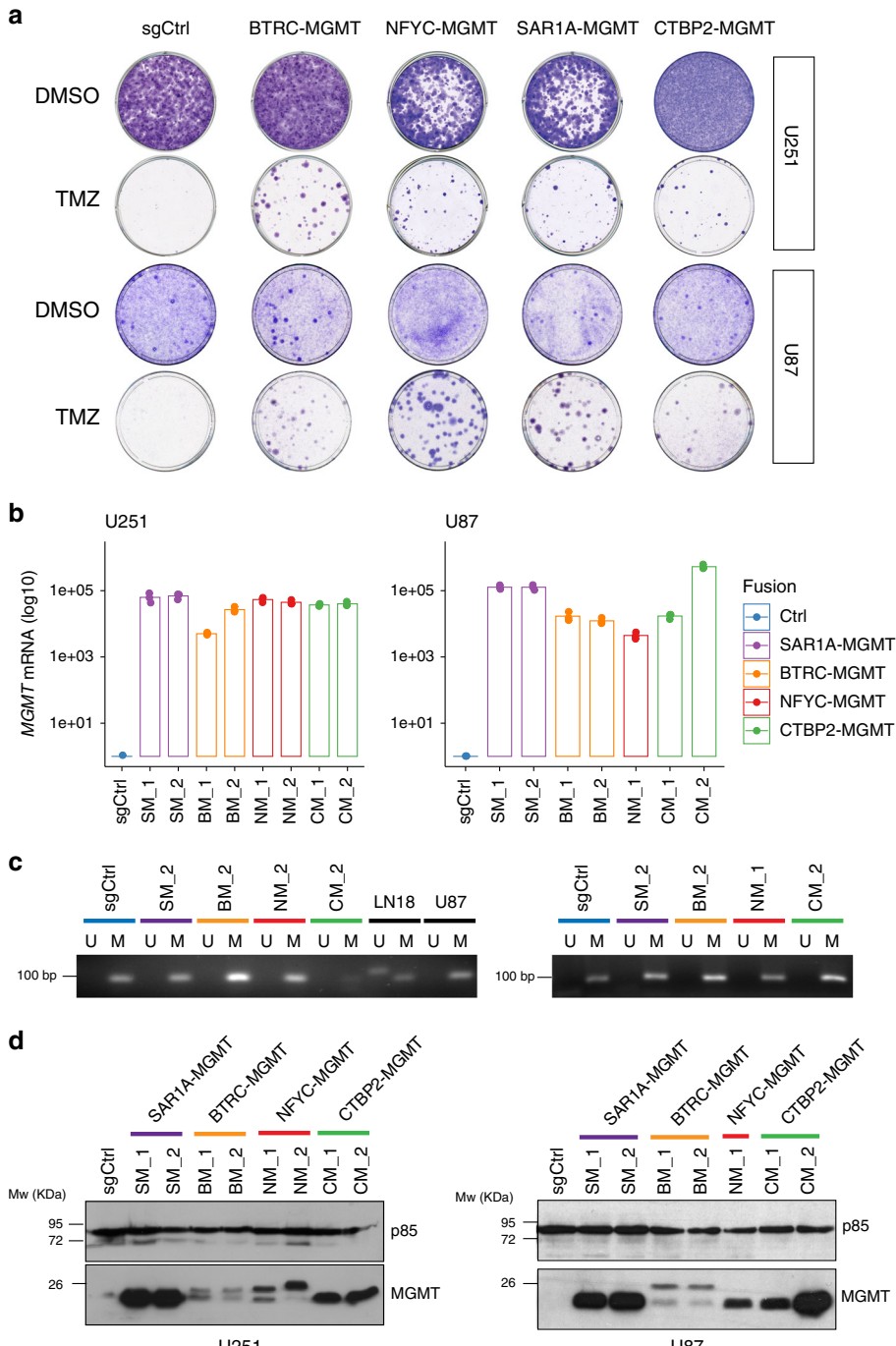

**Fig. 2 *MGMT* fusions cells show enhanced TMZ resistance via increased MGMT expression. a** Colony-forming assay performed on U251 and U87 cells expressing sgCtrl, BTRC-MGMT, NFYC-MGMT, SAR1A-MGMT, CTBP2-MGMT exposed for 12 days to TMZ (100 μM) or DMSO. **b** MGMT quantitative-PCRs performed on mRNA from U251 and U87 TMZ-resistant single-cell clones expressing the indicated MGMT fusions. Data are from a representative experiment of $n = 3$ biological replicate. Centre of the bars represent the mean (technical replicate $n = 3$) and the error bars are the standard deviations. **c** Analysis of *MGMT* promoter methylation, by methylation-specific PCR (MSP), in the TMZ-resistant cell clones expressing the indicated *MGMT* fusions from U251 (left panel) and U87 (right panel). M and U lanes indicate methylated and unmethylated status of the promoter, respectively. LN18 and U87 cells are shown as control for unmethylated and methylated, respectively. **d** Western blot analysis of MGMT protein levels in TMZ-resistant cell clones from U251 and U87 expressing the indicated *MGMT* fusions. Source data are provided as a Source Data file.

genotype, phenotype, as well as heterogeneity of the original tumor both in vitro and in vivo. We generated and confirmed at genomic and mRNA levels the *BTRC-MGMT* and *SAR1A-MGMT* fusion events in the MGMT-negative patient-derived h543 GBM tumor spheres (Supplementary Fig. 7a–c). Similar to what observed for U251 and U87, h543 cells carrying the fusions

showed increased expression of MGMT at mRNA and protein level, as compared with control cells (Supplementary Fig. 7d, e).

**MGMT gene fusions contribute to TMZ resistance.** To establish whether the TMZ resistance in the clones carrying the fusions was determined by the overexpression of a fully functional MGMT

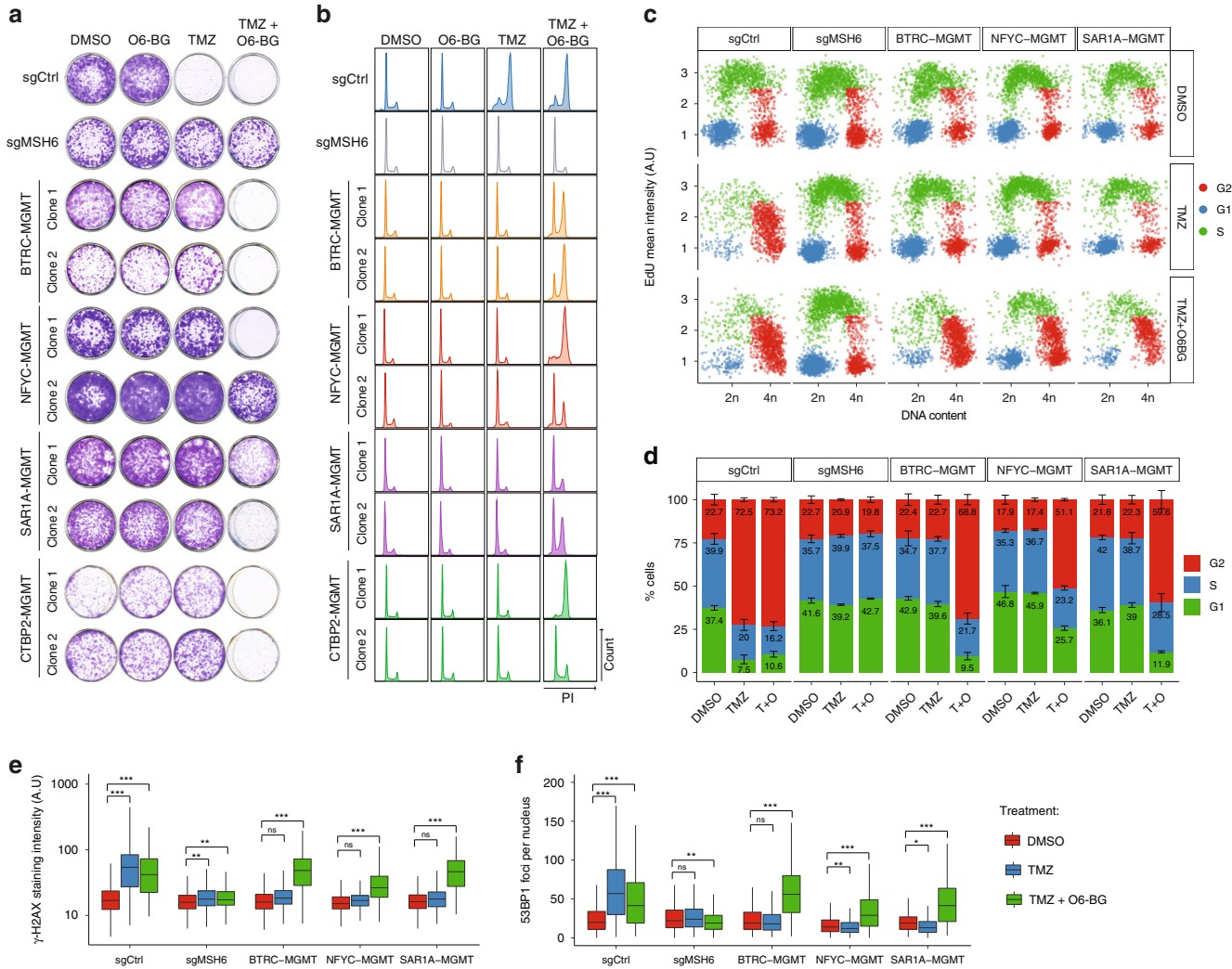

**Fig. 3 MGMT fusions protect from TMZ-induced damage. a** Clonogenic survival assay of U251 clones expressing MGMT fusions exposed to O6-BG (100 μM) or/and TMZ (100 μM) for 12 days. U251sgMSH6 cells are shown as control for TMZ resistance independently from MGMT. **b** Cell-cycle distribution of U251 MGMT fusion expressing cells in presence of O6-BG (100 μM) or/and TMZ (100 μM) for 72 h, measured by propidium iodide (PI) staining and FACS. U251 sgCtrl and sgMSH6 are shown as control. **c** High-throughput microscopy-mediated quantification of cell-cycle distribution at 48 h after treatment. See "Methods" for details. **d** Quantification of the percentage of cells in **c**. Data are from a representative experiment repeated in triplicate and presented as mean (technical replicate $n = 3$) and standard deviation. **e, f** High-throughput microscopy-mediated quantification of γH2AX intensity levels and 53BP1 foci in U251 cells expressing the *MGMT* fusions after 48 h of treatment with 100 μM of the indicated drugs. U251 sgCtrl and sgMSH6 were included as controls. The bottom and top of each box represents the first and third quartiles, and the line inside is the median. The whiskers correspond to 1.5 times the interquartile range. Data are representative of $n = 3$ biologically independent experiments. Two-sided Student's *t* test with Bonferroni adjustment for multiple comparisons: ***$P < 0.001$, **$P < 0.01$, *$P < 0.05$, ns not significant, A.U. arbitrary unit. Source data are provided as a Source Data file.

protein, and not caused by other mutations acquired during TMZ treatment, we analyzed the TMZ sensitivity in presence of O6-benzylguanine (O6-BG), a synthetic derivative of guanine that inhibit MGMT activity[15]. Clonogenic assay of two independent U251 clones per fusion showed that the TMZ sensitivity was re-established by the co-treatment with O6-BG (Fig. 3a). By contrast, cell knockout for the mismatch repair gene MSH6, a proposed TMZ-resistance mechanism independent from MGMT expression, was fully TMZ-resistant also in the presence of O6-BG. Similarly, cell-cycle profile analysis with propidium iodide staining and EdU incorporation assays showed that the fusion clones bypassed the TMZ-induced accumulation in the G2/M phase and O6-BG co-treatment was able to re-establish the cell-cycle arrest (Fig. 3b–d). We noticed that individual clones showed variable

TMZ sensitivity when treated concurrently with O6-BG. Clones with higher MGMT expression (e.g., NFYC-MGMT clone 2 and SAR1A-MGMT clones) showed increased resistance to TMZ; however, in these cells increasing doses of O6-BG significantly enhanced TMZ cytotoxic effect (Supplementary Fig. 5a, b). Same results were obtained in U87 fusion clones (Supplementary Fig. 6b, c) and in h543 tumor spheres (Supplementary Fig. 7f, g).

We then assessed to which extent the TMZ resistance was determined by increased MGMT activity, and therefore boosted by the DNA repair potential of the fusion clones. Quantitative high-throughput microscopy analysis revealed that in MGMT fusion expressing cells, similarly to what observed in sgRNA MSH6 cells, TMZ treatment did not increase levels of γH2AX and 53BP1 foci, DNA damage markers characteristic of cells bearing

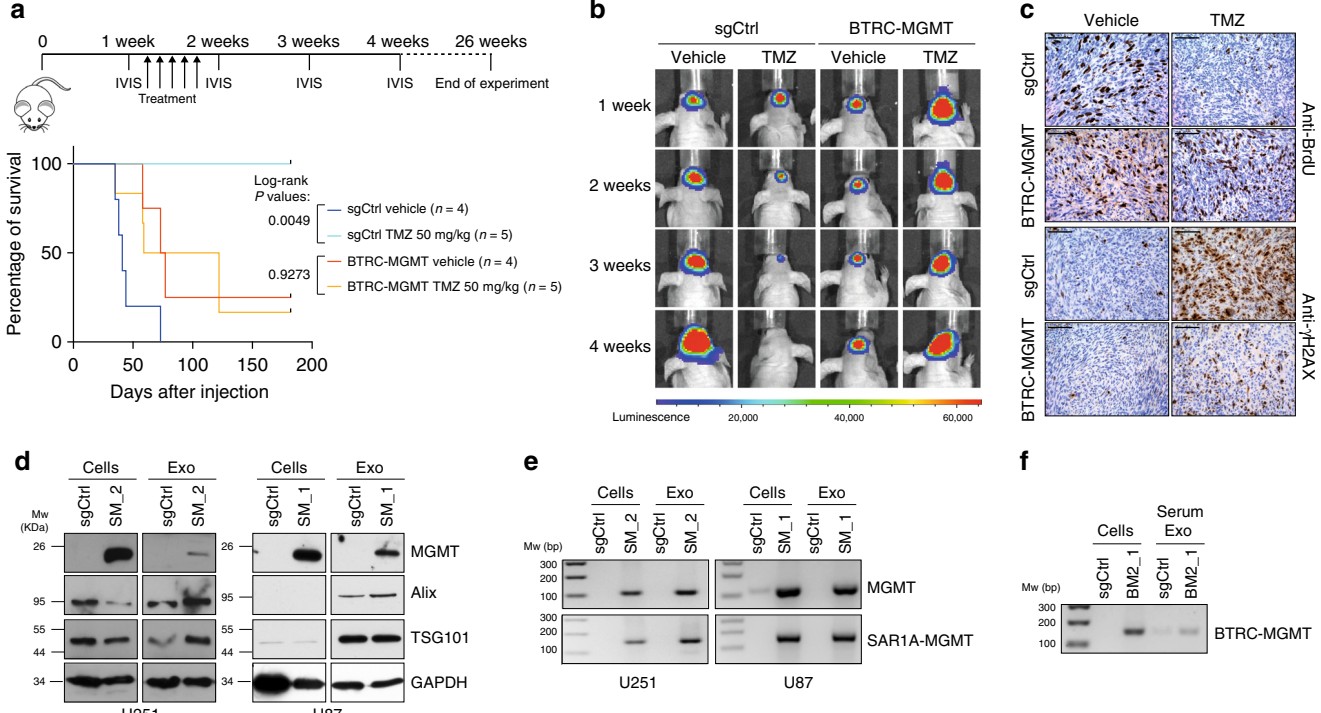

**Fig. 4 MGMT fusions confer TMZ resistance in vivo and serve as biomarkers at recurrence. a** Top panel: scheme of the in vivo experimental design. Bottom panel: Kaplan–Meier survival curve of animals intracranially injected with U251 sgCtrl and U251 BTRC-MGMT clone 2 cells transduced with a luciferase construct, treated or not with TMZ (50 mg/Kg) for 5 days: sgCtrl Vehicle n = 4, sgCtrl TMZ n = 5, BTRC-MGMT vehicle n = 4, BTRC-MGMT TMZ n = 5. sgCtrl log-rank P-value = 0.0049, BTRC-MGMT log-rank P-value = 0.9273. **b** Representative luminescent images of the tumor bearing at the indicated time points. **c** Immunohistochemistry analysis against BrdU and γH2AX of tumors from mice injected with U251 sgCtrl and BTRC-MGMT clone 2 cells, treated or not with TMZ (50 mg/kg) for 3 days. Mice were sacrificed 2 h after BrdU injection. Scale bars: 100 μm. **d** Western blot analysis of the EXO markers Alix and TSG101 and of MGMT levels in samples pair of cells and cell-derived EXOs expressing sgCtrl and SAR1A-MGMT. **e** SAR1A-MGMT and MGMT mRNA expression by RT-PCR in RNA pair samples from cells and cell-derived EXOs expressing sgCtrl and SAR1A-MGMT. **f** Transcript levels of BTRC-MGMT by RT-PCR analysis in EXOs isolated from serum of BTRC-MGMT clone 2 tumor-bearing mice compared to sgCtrl mice. U251 sgCtrl and BTRC-MGMT clone 2 cells were included as controls. Source data are provided as a Source Data file.

DNA double-strand breaks (Fig. 3e, f). However, MGMT inhibition by O6-BG led to the accumulation of γH2AX and 53BP1 foci upon TMZ treatment in the fusion clones. Taken together, these data indicate that TMZ resistance induced by MGMT genomic rearrangements is mechanistically linked to MGMT activity.

**MGMT gene fusions protect from TMZ treatment in vivo.**
Lastly, we evaluated the TMZ resistance of MGMT fusion in vivo through establishing nu/nu mice xenograft models with the U251 BTRC-MGMT and control cells, previously transduced with a luciferase expressing construct. A week after intracranial transplantation, mice were intraperitoneally treated with TMZ (50 mg/Kg) or DMSO (0.3%) for 5 days, and tumor growth was monitored weekly with bioluminescence imaging (BLI) for 4 weeks. Mice with MGMT fusion-bearing tumors exhibited no significant prolonged lifespan between TMZ and DMSO group, and significantly poorer survival compared with control mice when receiving TMZ treatment (Fig. 4a). Similarly, while TMZ treatment significantly extended the survival of mice transplanted with h543 transduced with the sgCtrl, it failed to do so in those expressing the SAR1A-MGMT rearrangement (Supplementary Fig. 7h). BLI analysis confirmed that TMZ antitumor effect was limited to control mice (Fig. 4b). In addition, as shown by immunohistochemistry, the BTRC-MGMT mice had increased BrdU incorporation and reduced accumulation of γH2AX

compared with control mice upon TMZ administration (Fig. 4c), confirming our proliferation and DNA repair in vitro results.

In clinical settings, liquid biopsies can be a powerful noninvasive technique to monitor cancer-associated genetic alterations by analyzing circulating tumor cells (CTCs), circulating free DNA (cfDNA) or tumor-derived extracellular vesicle (EV), including exosomes (EXOs). Previous studies have already showed that (i) glioma-derived extracellular vesicle (EV) can cross the blood brain barrier and be detected in peripheral blood of patients[16], (ii) MGMT mRNA is enriched in glioma exosomes (EXOs)[17], and (iii) other gene fusion was identified in glioma EXOs[18]. Based on these findings, we assessed whether the MGMT fusions could be detected in EXOs. We purified EXOs from conditioned media of cells harboring SAR1A-MGMT and sgCtrl by standard ultracentrifugation. Western blot of protein content confirmed enrichment in the EXOs of the exosome-specific markers TSG101 and Alix (Fig. 4d) and the presence of MGMT in the cells expressing the fusion event (Fig. 4d). Most importantly, also the mRNA of the MGMT fusion was detected by RT-PCR in the EXOs (Fig. 4e).

Lastly, to further evaluate a clinical application of our findings, we tested whether EXOs isolated from blood serum of mice injected orthotopically with the U251 BTRC-MGMT cells would also exhibit the fusion transcript. Remarkably, RT-PCR analysis confirmed the presence of the cDNA fusion fragment in the BTRC-MGMT-derived circulating blood EXOs (Fig. 4f).

## Discussion

Currently, TMZ is the only chemotherapeutic drug that is established to considerably extend the overall survival of GBM patients and is becoming a therapeutic option also for high-risk low-grade gliomas[9]. Both intrinsic and acquired resistance might contribute to glioma tumor recurrence upon TMZ treatment. While *MGMT* promoter hypomethylation is undoubtedly recognized as the primary mechanism of intrinsic TMZ resistance, the genetic alterations acquired during TMZ exposure that contribute to tumor relapse still remain to be fully characterized.

Defects in various components of the MMR machinery possibly represent one of the most well-characterized mechanism of acquired TMZ resistance. Though rarely detected in primary GBMs, MMR alterations have been previously described in 10–20% of recurrent tumors[7,19,20]. Changes in *MGMT* promoter methylation status during tumor progression have been observed only in a small subset of patients[19]. More recently, it has also been suggested that in recurrent GBMs enhancer hijacking could promote MGMT expression, despite promoter methylation, and therefore TMZ resistance; however, the clinical significance of these findings still remain to be evaluated[5].

In this study, we demonstrated that *MGMT* fusions represent a previously unidentified genetic alteration that contribute to MGMT overexpression and a novel mechanism of acquired TMZ resistance that is mutually exclusive from *MGMT* promoter hypomethylation and the hypermutator phenotype, typically associated with MMR defects. For those patients for whom both primary and recurrent tumor were available (4 out of 7), the *MGMT* rearrangements were detected only in the tumor relapse. Although we cannot exclude that some of the primary tumors might express the *MGMT* fusion at subclonal level, and therefore possibly lower than the RNA-seq detection limits, we speculate that the *MGMT* rearrangements have been acquired during the course of TMZ treatment and then positively selected due to their ability of driving TMZ resistance. Very recently, another *MGMT* gene fusion, *ASAP2-MGMT*, with similar features to the fusions that we have described here in gliomas, has been identified in a medulloblastoma patient that relapsed after TMZ treatment[21]. These data would suggest that *MGMT* genomic rearrangements could represent a relevant mechanism of resistance to alkylating agents across a broader spectrum of tumor types.

Although the presence of the *MGMT* gene fusions in extracellular vesicles appears to be promising as a possible liquid biopsy approach for the identification of *MGMT* rearrangements, its validity still remains to be validated in the clinical settings. Early detection of *MGMT* genetic rearrangements in patients under treatment would eventually predict early tumor recurrence and guide therapy decision in a subset of MGMT-methylated patients. Unlike primary tumors, at the time of recurrence there is not a standard of care available for gliomas, and TMZ rechallenge is one of the few options in glioblastomas[22]. *MGMT* promoter methylation has been proposed as prognostic marker for benefit from TMZ rechallenge in recurrent glioblastoma[23] and is used as a stratification factor in trials comprising TMZ treatment[24]. However, our current findings might limit *MGMT* promoter methylation prognostic value and would predict that a subset of patients might be assigned to the wrong treatment arm, if based solely on *MGMT* promoter methylation analysis.

In summary, here we have presented *MGMT* genomic rearrangements not only as a novel mechanism of resistance to TMZ in a subset of gliomas but also, to our knowledge, as a unique genetic alteration never described before in response to other chemotherapeutic agents.

## Methods

**Patients**. The newly sequenced tumors were collected from Beijing Tiantan Hospital as part of the Chinese Glioma Genome Atlas project (CGGA, http://cgga.org.cn/). The study was approved by the institutional review board in Capital Medical University (IRB ID: KYSB2015-023). Informed consent was obtained from each patient before surgery. For each specimen, the pathological diagnosis was reviewed by board-certificated pathologists. The specimen was flash-frozen within 5 min after being resected for subsequent RNA extraction and sequencing.

We also curated RNA sequencing from four published studies. This include 72 samples from Wang et al.[7], 42 samples from Hu et al.[25], 28 samples from Bao et al.[26], and 5 samples from The Cancer Genome Atlas[27] (Supplementary Fig. 1 and Supplementary Data 1).

The most recent follow-up information of the TCGA patients were retrieved from NCI Genomics Data Commons (GDC) data portal (https://portal.gdc.cancer.gov, accessed on July 18, 2019). Similarly, we used the most recent follow-up information (last follow-up in December 2018) of all patients from CGGA. For the 41 patients from Samsung Medical Center (SMC), patient follow-up continued after the publication of our last study[7], and the updated data were used in this study. In total, 12 out of 41 patients changed survival status and/or surviving time. In addition, the MGMT methylation status of the recurrent gliomas from seven patients were newly tested and updated in this study.

**RNA sequencing and gene expression quantification**. RNA-sequencing assay of the newly collected glioma samples in this study was performed using the same protocol as our previous research[28]. For each sample, about 80 million reads were generated.

The cleaned RNA-sequencing reads were mapped to the reference human genome assembly of Ensembl GRCh37 annotation version 75 using STAR 2.6.1d[29] with default parameters. Reads mapped to each gene were counted using FeatureCount 1.5.1[30] and transformed to RPKM. Since our cohort includes samples from multiple cohorts, we used Z-score of MGMT expression in the recurrent glioma samples within each cohort for normalization to overcome potential batch effects.

**Detection of *MGMT* fusion from RNA-sequencing data**. RNA-sequencing data from previous publications were downloaded, and the reads were extracted using samtools 1.2[31]. STAR-fusion 1.5.0[32] was utilized to identify and annotate gene fusion candidates, using the fastq files as input. The fusion candidates were then filtered by removing fusions that were present in normal tissues, fusions involving mitochondria genes and uncharacterized genes, and fusions of two paralog genes. See Supplementary Table 1 for the breakpoint information of the MGMT fusions identified.

**Whole-genome sequencing and analysis**. For one *MGMT* fusion positive case (CGGA_1729), we had enough sample for whole-genome sequencing. The total DNA was extracted and sequenced using Illumina HiSeq 4000 platform. The sequencing depth is about 50×. Sequencing reads were then cleaned and mapped to hg19 reference genome using bwa mem 0.7.15-r1140[33]. Duplicates were marked using Picard MarkDuplicates 2.9.2 tool (https://broadinstitute.github.io/picard/). Structural variants were identified using Manta 1.4.0[34], and the variant related to the *MGMT* fusion was manually picked.

**Determination of *IDH*, 1p/19q co-deletion, and hypermutation**. The mutation status of IDH1 Arginine 132 and IDH2 Arginine 172 were determined from RNA-seq data using samtools 1.2 mpileup. At least five reads were required to cover the hotspot position, otherwise the result was marked as not available (NA).

The 1p/19q co-deletion status was predicted using CNAPE[35]. CNAPE is a software to predict large-scale copy number alteration from gene expression data using multinomial logistic regression models trained on TCGA data and have shown high sensitivity and specificity. The 1p/19q co-deletion prediction results were further confirmed by the allele frequency of common SNPs. Hypermutation was identified using a computational method based on RNA-sequencing data[36].

**A bootstrapping method to test mutual-exclusiveness**. To test whether the three TMZ-resistance-related alterations, namely *MGMT* promoter hypomethylation, hypermutation, and *MGMT* fusion, are mutually exclusive, we reasoned that if they are mutual exclusive, then when combined they should cover significantly more patients than random. Note the contraposition also holds. We therefore randomly assigned the patients whether they had the alteration and summarized the number patients that had at least one of the three alterations. This randomized assignment was repeated for 10,000 times. *P*-value was calculated by (times for which the number of covered patients is larger than the observed number of patients carrying at least one such alteration)/10,000.

**PCR validation of MGMT fusion in patient samples**. The total RNA was extracted from the positive fusion glioma samples using RNeasy Mini Kit (Qiagen) according to the manufacturer's instructions, and RNA intensity was examined by Bioanalyzer 2100 (Agilent Technologies). Then cDNA was synthesized from 1 µg of

the total RNA using the RevertAid First Strand cDNA Synthesis kit (Thermo Fisher Scientific, Cat. K1622), with random hexamer as the primer. The MGMT fusion gene fragments were amplified by PCR using specific primers (Supplementary Table 2). The PCR products were purified using a QIAquick PCR purification kit (Qiagen, Cat. 28104) and sequenced by an ABI Prism 3730 DNA sequencer (Applied Biosystems).

**DNA constructs, design, and cloning of guide RNAs.** The pKLV-U6gRNA-PGKpuro2ABFP (Plasmid #50946) and the lentiCas9-Blast (Plasmid #52962) were obtained from Addgene. The HSV1-tk/GFP/firefly luciferase (TGL) triple-reporter construct was from J. Gelovani Tjuvajev[37]. The gRNA sequences targeting MGMT, BTRC, NFYC, SAR1A, and CTBP2 were designed using the Genetic Perturbation Platform web portal (http://portals.broadinstitute.org/gpp/public/analysis-tools/gRNA-design) (Supplementary Table 3). The paired sgRNAs were sub-cloned into the pKLV-U6gRNA-PGKpuro2ABFP, as previously described[11]. Briefly, the oligonucleotides containing the different gRNA pairs (Supplementary Table 4) were amplified with Phusion High-Fidelity polymerase (New England Biolabs, M0530S) using primer F5 and R1 (Supplementary Table 2). PCR products were gel-purified and ligated to BbsI-digested pDonor_mU6 plasmid (kindly provided by A. Ventura) by using the Gibson Assembly Master Mix (New England Biolabs 174E2611S). The Gibson reaction was then digested with BbsI at 37 °C for 3 h. The linearized fragment containing the pair gRNA, the mU6 promoter, and the gRNA scaffold was gel-purified and cloned into the pKLV-U6gRNA-PGKpuro2ABFP. All the constructs were verified by Sanger sequencing.

**Cell lines, transfections, infections, and reagents.** The human glioma cell lines U251 (Sigma-Aldrich, 09063001) was kindly provided by Eric Holland, and U87 (HTB-14) was purchased from ATCC. The Gp2-293 packaging cell line was purchased from Clontech (Cat. 631458). Cells were cultured in DMEM (Sigma-Aldrich, Cat. D5796) + 10% FBS (Sigma-Aldrich, Cat. F7524). All the cell lines were routinely checked for Mycoplasma contamination by PCR analysis. DNA fingerprinting has been performed for authentication of the U251 and U87 cell lines (data available upon request). Human GBM tumor spheres h543, kindly provided by Eric Holland, were cultured in human NeuroCult NS-A Proliferation Kit (Stem Cell Technologies, Cat. 05751) and supplemented with 10 ng/ml recombinant human EGF (Gibco, Cat. PHG0313), 20 ng/ml basic-FGF (Sigma-Aldrich, Cat. F0291-25UG), 1 mg/ml heparin (Stem Cell Technologies, Cat. 07980), 100 U/ml penicillin, and 100 μg/ml streptomycin.

Lentiviruses were generated by co-transfection of lentiviral plasmids (pKLV-U6gRNA-PGKpuro2ABFP and lentiCas9-Blast) and 2nd generation packaging vectors (pMD2G and psPAX2) in Gp2-293 cells using calcium–phosphate precipitate. High-titer virus was collected at 36 and 60 h following transfection, and used to infect cells in presence of 7 μg/ml polybrene (Sigma-Aldrich, Cat. H9268-5G) for 12 h. Transduced cells were selected with Blasticidin (3 μg/ml) (Gibco, Cat. A11139-03) and Puromycin (1.5 μg/ml) (Sigma-Aldrich, Cat. P8833-25MG). To isolate clones of U251 and U87 cells carrying the desired MGMT genomic rearrangements, after transduction with the specific pKLV-dual gRNA vectors, the cells were selected with Puromycin and then exposed to Temozolomide. Single TMZ-resistance clones were then recovered with cloning cylinders and then expanded.

Temozolomide was purchased from Selleckchem (Cat. S1237). $O^6$-benzylguanine was from Sigma-Aldrich (Cat. B2292-50MG).

**Immunoblotting.** Cells were lysed with RIPA lysis buffer (20 mM Tris-HCl, 150 mM NaCl, 1% NP-40, 1 mM EDTA, 1 mM EGTA, 1% sodium deoxycholate, 0.1% SDS), and protein concentrations were determined by DC protein assay kit (Biorad, Cat. 5000111). Proteins were run on house-made SDS-PAGE gels and transferred to nitrocellulose membrane (Amersham, Cat. GEHE10600003). Membranes were first incubated in blocking buffer (5% milk 0.1% Tween, 10 mM Tris at pH 7.6, 100 mM NaCl) and then with primary antibody MGMT (Biosciences, Cat. 557045, Lot. 6280927, 1:2000), Alix (Cell Signaling, Cat. 2171, Lot. 5, 1:1000), TSG101 (BD Transduction Laboratories, Cat. 612696, Lot. 7208980, 1:2000) overnight at 4 °C, p85 (Millipore, Cat. 0619, Lot. 3009962, 1:10,000), GAPDH (Santa Cruz, Cat. Sc-365062, Lot. J1314, 1:500), and Vinculin (Sigma-Aldrich, Cat. V9131, 1:10.000) for 1 h at room temperature. Anti-mouse or rabbit-HRP conjugated antibodies (Jackson Immunoresearch) were used to detect desired protein by chemiluminescence with ECL (Amersham, RPN2106).

**Immunohistochemistry.** Tissue samples were fixed in 10% formalin, paraffin-embedded and cut in 3-μm sections, which were mounted in superfrostplus microscope slides (Thermo Scientific, Cat. 165061) and dried. The immunohistochemistry was performed using an automated immunostaining platform (Ventana discovery XT, Bond Max II, Leica). Antigen retrieval was performed with low pH buffer (CC1m) for p-H2AX and high pH buffer (ER2) for BrdU. Endogenous peroxidase was blocked (peroxide hydrogen at 3%), and slides were then incubated with anti-BrdU (BU-1, GE Healthcare, RPN202, Lot. 341585, 1:100) and phospho-histone H2AX (Ser139) (γH2AX, JBW301, Millipore, 05-636, Lot. DAM1493341, 1:4000). After the primary antibody, slides were incubated with the corresponding secondary antibodies when needed (rabbit anti-mouse Abcam) and visualization

systems (Omni Map anti-Rabbit, Ventana, Roche; Bond Polymer Refine Detection, Bond, Leica) conjugated with horseradish peroxidase. Immunohistochemical reaction was developed using 3,30- diaminobenzidine tetrahydrochloride (DAB) (ChromoMap DAB, Ventana, Roche; Bond Polymer Refine Detection, Bond, Leica), and nuclei were counterstained with Carazzi's hematoxylin. Finally, the slides were dehydrated, cleared, and mounted with a permanent mounting medium for microscopic evaluation.

**Colony-forming assay.** Cells were seeded in six-well culture plates (5000 per well) or in 12-well plates (2200 per well) in triplicate. After 4 h from the seeding, Temozolomide (100 or 200 μM) and/or $O^6$-benzylguanine (100 μM) were added to the cells, and fresh media with drugs was replaced after 6 days. Twelve days after plating, resistant colonies were either stained with 0.5 M of crystal violet (Alfa Aesar, Cat. B21932) or isolated using cloning cylinders (Corning, Cat. 31666) and subsequently amplified.

**Flow cytometry.** Cells were seeded in six-well culture plates (100,000 per well) in duplicates and cultured in presence of temozolomide (100 μM) and/or $O^6$-benzylguanine (100 μM) for 72 h. Cells were then harvested by phosphate-buffered saline (PBS), washed twice in cold PBS, fixed with cold 100% ethanol on ice for 30 min, and pelleted by centrifugation at 1200 rpm for 10 min. Pellet was then washed twice with PBS and 1% fetal bovine serum (FBS) and stained with 200 μl of propidium iodide (PI) (50 μg/ml) overnight. Samples were acquired on a FACS Canto II (Beckton Dickinson). All data were analyzed using FlowJo 9.9.4 (Treestar, Oregon). Gating strategy is described in Supplementary Fig. 8.

**MTT assays.** For viability assays, $10^4$ tumor spheres h543 were plated per well in 96-well plates. After addition of the indicated concentrations of temozolomide or vehicle, cells were incubated for 7 days at 37 °C and 5% CO₂. In total, 10 μl of MTT reagent (Sigma, 5 mg/ml in PBS) were then added to the media and incubated for 4 h. After adding 100 μl of a 1% SDS, 4 mM HCl solution, absorbance at 595 nm was recorded after 24 h with a plate reader.

**High-throughput microscopy.** Cells (2000 per well) were grown on a μCLEAR bottom 96-well plates (Greiner Bio-One, Cat. 736-0230) and treated with temozolomide (100 μM) and/or $O^6$-benzylguanine (100 μM) in triplicates for 48 h. EdU (10 μM) (Life Technologies, S.A., Cat. A10044) was added to the media at the last hour of incubation with the drugs. Cells were then fixed in 4% PFA for 20 min, permeabilized, and incubated for 1 h in blocking solution (3% BSA in 0.1% Triton-X PBS). EdU incorporation was detected using the Click-iT™ EdU Alexa Fluor® Imaging kit (Life Technology, S.A., Cat. C-10425). Phospho-histone H2AX (Ser139) (γH2AX, Merck, Cat. 05-363, Lot. 2310355, 1:1000) and 53BP1 (Novus Biologicals, Cat. NB100-304, Lot. A2, 1:3000) immunofluorescence was performed using standard procedures. Cells were incubated with primary antibodies overnight at 4 °C, and secondary antibodies conjugated with Alexa 488 (rabbit) (Life Technologies, Cat. A-21206, Lot. 198155, 1:400) or Alexa 555 (mouse) (Life Technologies, Cat. A-31570, Lot. 1048568, 1:400). Nuclei were visualized by DAPI staining (Sigma-Aldrich, Cat. D8417). Images from each well were automatically acquired by an Opera High-Content Screening System (Perkin Elmer) at non-saturating conditions with a ×20 (γH2AX) and ×40 (53BP1) magnification lens. Images were segmented using the DAPI staining to generate masks matching cell nuclei from which the mean signals were calculated. Cell-cycle phases were inferred based on DNA content (DAPI intensity*nuclear area) and EdU mean intensity: cells with 2n DNA content and EdU-negative were considered as G1 phase; <4n DNA content and EdU-positive, as S phase; 4n DNA content and EdU low or negative, as G2 phase.

**Genomic DNA isolation, gene fusion analysis, and methylation-specific PCR.** Genomic DNA was isolated as previously described[11]. Briefly, cell pellets were incubated in lysis buffer (10 mM Tris-HCl ph8, 100 mM NaCl, 0.5 mM EDTA, 10% SDS, and proteinase K) for 4 h at 55 °C, and genomic DNA was extracted using phenol:chloroform (1:1) and Phase Lock heavy 2-ml tubes (5PRIME, Cat. 2302830). In all, 0.1 M sodium acetate and 100% cold ethanol were then added to the recovered aqueous phase. Samples were centrifuged at 15,000 rpm for 25 min. After washing in 70% cold ethanol, draining, and dissolving in water, genomic DNA was quantified.

For detection of gene fusion events, 100 ng of DNA were amplified with specific primers listed in (Supplementary Table 2). PCR products were cloned into the pGEM-T Easy vector (Promega, Cat. A1360) and submitted to Sanger sequencing.

The MGMT promoter methylation status was determined by methylation-specific PCR (MSP). In total, 2 μg of DNA were subjected to bisulfite treatment using the EpiTect® Bisulfite kit (Quiagen, Cat. 59104). DNA was cleaned up following the manufacturer's instructions and quantified. In all, 30 ng of DNA per sample were PCR-amplified with the Platinum SuperFi DNA polymerase (Invitrogen, Cat. 12351-010) and specific primers to detect methylated and unmethylated MGMT promoter (Supplementary Table 2). The PCR amplification protocol was as follows: 94 °C for 1 min, then denature at 94 °C for 30 s, anneal at 60 °C for 30 s, extension at 70 °C for 30 s for 35 cycles, followed by a 7-min final extension.

**Reverse transcription quantitative PCR and analysis of cDNA fragments**. RNA from cells was isolated with TRIzol reagent (Invitrogen, Cat. 15596-026) according to the manufacturer's instructions. For reverse transcription PCR (RT-PCR), 500 ng of the total RNA was reverse transcribed using the High Capacity cDNA Reverse Transcription Kit (Applied Biosystems, Cat. 4368814). The cDNA was used either for quantitative PCR or Sanger sequencing. The cDNA was PCR-amplified using primers listed in Supplementary Table 2, in-gel-purified and ligated into the pGEM-T Easy vector (Promega, Cat. A1360) and submitted to Sanger sequencing. Quantitative PCR was performed using the SYBR-Select Master Mix (Applied Biosystems, Cat. 4472908) according to the manufacturer's instructions. qPCRs were run, and the melting curves of the amplified products were used to determine the specificity of the amplification. The threshold cycle number for the genes analyzed was normalized to ACTIN. Sequences of the primers used are listed in Supplementary Table 2.

**Fluorescence in situ hybridization (FISH)**. Two sets of FISH probes were used to study the various MGMT genomic rearrangements. Bacterial artificial chromosomes (BACs) that map at the 5′ and 3′ MGMT flanking regions (10q26 cytoband) were purchased from BACPAC Resoirce CHORI and labeled by Nick translation assay with Spectrum Green (RP11-165L12 and RP11-343L20) and Spectrum Orange (RP11-960B17 and RP11-357N5) fluorochromes, respectively, to generate a break-apart locus-specific FISH probe. FISH analyses were performed according to the manufacturers' instructions, on Carnoy's fixed cells mounted on positively charged slides (SuperFrost, Thermo Scientific). Briefly, the slides were first dehydrated followed by a denaturing step in the presence of the FISH probe at 85 °C for 10 min and left overnight for hybridization at 45 °C in a DAKO hybridizer machine. Finally, the slides were washed with 20 × SSC (saline-sodium citrate) buffer with detergent Tween-20 at 63 °C, and mounted in fluorescence mounting medium (DAPI). FISH signals were manually enumerated within nuclei. FISH images were also captured using a CCD camera (Photometrics SenSys camera) connected to a PC running the CytoVision Version 7.4 image analysis system (Applied Imaging Ltd., UK).

**Exosomes isolation**. To purify exosomes from cell culture, the conditioned media was collected after 72 h from 10 × 15 cm plates and centrifuged at 500 × g for 10 min followed by centrifugation at 12,500 × g for 25 min and 100,000 × g for 80 min. The exosome pellet was then washed with cold PBS, centrifuged at 100,000 × g for 80 min and re-suspended in 100 μl PBS. Isolation of exosomes from mice serum was performed following the same protocol after an initial centrifugation at 3000 × g for 20 min and a further one at 12,000 for 20 min. NanoSight analysis was used to confirm the integrity and expected size of the isolated exosomes. Centrifugations were done at 10 °C using a Beckman Optima X100 ultracentrifuge with a Beckman 50.4Ti or 70.1Ti rotor. Exosome protein content was determined by DC protein assay kit. Particle content was determined by measuring 1 μl of exosome aliquot diluted in 1 ml PBS with an NTA (NanoSight; Malvern) equipped with a blue laser (405 nm).

**Orthotopic GBM models, bioluminescence imaging, and in vivo treatment**. U251 sgCtrl and BTRC-MGMT cells were stably transduced with the HSV1-tk/GFP/firefly luciferase (TGL) triple-reporter construct and GFP positive cells were purified by FACS. Four to five weeks old immunodeficient nu/nu mice were then intracranially injected with the sorted cells ($5 \times 10^5$ cells) using a stereotactic apparatus (Stoelting). After intracranial injection, mice were imaged every week to follow tumor growth and drug response. Mice were anesthetized with 3% isoflurane before retro-orbital injection with d-luciferin (150 mg/Kg) (Perkin Elmer S.L., Cat. 122796) and imaged with an IVIS Xenogen machine (Caliper Life Sciences). Bioluminescence analysis was performed using Living Image software, version 3. Beginning the day in which tumors were clearly visible by IVIS, mice were randomized into two groups, and temozolomide (50 mg/Kg) or vehicle (DMSO) was administered intraperitoneally daily for 5 days. For survival curve, mice were then checked until they developed symptoms of disease (lethargy, weight loss, macrocephaly). For IHC analysis, BrdU (150 μg) (Sigma-Aldrich, Cat. B9285) was administrated intraperitoneally to mice, and mice were then sacrificed 2 h later. For the h543 orthotopic model, $3 \times 10^5$ sgCtrl and SAR1A-MGMT were transplanted intracranially, and after 1 week mice were randomized in two groups, and temozolomide (100 mg/Kg) or vehicle (DMSO) was administered intraperitoneally daily for 5 days.

Mice were housed at 22 °C with a 12-h light/ 12-h dark cycle, in the specific pathogen-free animal house of the Spanish National Cancer Centre under conditions in accordance with the recommendations of the Federation of European Laboratory Animal Science Associations (FELASA). All animal experiments were approved by the Ethical Committee (CEIyBA) and performed in accordance with the guidelines stated in the International Guiding Principles for Biomedical Research Involving Animals, developed by the Council for International Organizations of Medical Sciences (CIOMS).

**Statistics and reproducibility**. Data in bar graphs are presented as mean and SD, except otherwise indicated. Results were analyzed by unpaired two-tailed Student's t tests or Wilcoxon rank-sum test using the R programming language (version 3.5.3).

Kaplan–Meier survival curves were produced either with GraphPad Prism 6 (Fig. 4a) or with the "survival" 2.44–1.1 and "survminer" 0.4.6 R packages (Supplementary Figs. 1e, f and 7g); P-values were generated using the log-rank statistic. Heatmap, boxplots, and barplots were made with the "ComplexHeatmap" 1.2.0, "ggplot2" 3.2.1 and "ggpubr" 0.2.5R packages, respectively. Each experiment was repeated independently (minimum n = 2), unless specifically indicated, with similar results.

**Reporting summary**. Further information on research design is available in the Nature Research Reporting Summary linked to this article.

## Data availability

The raw sequencing data of the newly sequenced samples are deposited in the European Genome-phenome Archive (EGA) (https://ega-archive.org/), accession study ID: EGAS00001004544, and in the Genome Sequence Archive (GSA) of the Beijing Institute of Genomics (BIG) Data Center Chinese Academy of Sciences (https://bigd.big.ac.cn), accession number BioProject ID: PRJCA001580. Data from SMC were available in EGA, accession study ID: EGAS00001001800. Data from TCGA were downloaded from NCI Genomics Data Commons (GDC) data portal (https://portal.gdc.cancer.gov). Previously published CGGA data were available at the GSA BIG (https://bigd.big.ac.cn) under accession number BioProject ID: PRJCA001746 and PRJCA001747. The reference human genome hg19 is downloaded from http://hgdownload.cse.ucsc.edu/goldenpath/hg19/bigZips/hg19.fa.gz, while the genome annotation file is downloaded from ftp://ftp.ensembl.org/pub/release-75/gtf/homo_sapiens/Homo_sapiens.GRCh37.75.gtf.gz. All the other data supporting the findings of this study are available within the article and its information files and from the corresponding author upon reasonable request. Source data are provided with this paper.

## Code availability

The 1p/19q co-deletion status was predicted using the custom CNAPE software available at: https://github.com/WangLabHKUST/CNAPE.

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

## Acknowledgements

We would like to acknowledge Claudia Savini, Susana García, and Hector Peinado for the help and discussion on the exosome isolation and analysis. We thank Mª Carmen Martin Guijarro and Francisco José Moya for performing the FISH staining. We thank Manuel Perez and Gadea Mata for their assistance with the high-throughput microscopy analysis. We also thank Alvaro Ucero for the help with the isolation of the blood from the mice. We are very grateful to Anne Harttrampf and Lilianne Massade for sharing treatment information of the medulloblastoma patient, in which they identified the *ASAP2-MGMT* fusion. This research was supported by funds from the Seve Ballesteros Foundation and the Asociación Española Contra el Càncer (AECC) to M.S. This work was also supported by Natural Science Foundation of China (NSFC)/Research Grants Council (RGC) Joint Research Scheme (81761168038 to TJ and N_HKUST606/17 to J.W.), RGC-ECS grant 26102719, the NSFC grant (No. 31922088), ITC grant (ITCPD/17-9), Beijing Municipal Administration of Hospitals Clinical Medicine Development of Special Funding Support (ZYLX201708), Beijing Municipal Administration of Hospitals' Mission Plan (SML20180501), Beijing Nova Program (Z171100001117022) and Beijing Talents Foundation from Organization department of Municipal committee of the CPC (2017000021223ZK32).

## Author contributions

M.S. and J.W. conceived and supervised the study. T.J. provided patient samples. T.J., Z.B., and Z.W. contributed to patient follow-up, tissue collection, and sequencing of the CGGA cohort. J.S. and D.N. updated clinical data of the SMC cohort. B.O., M.S., Q.M., and J.W. wrote the paper. B.O. designed and performed experiments. N.V.-Q., P.K., Y.Z., H.L., and M.G.-G. performed experiments. Q.M. performed computational analysis. S.R.-P. designed and analyzed the FISH assays. J.Z., H.K., R.V., and R.R. helped with data analysis.

## Competing interests

The authors declare no competing interests.
