## [Peer Review File · Nature Communications]

Reviewers' comments:

Reviewer #1 (Remarks to the Author):

This manuscript describes a potential resistance mechanism in temozolomide-treated glioblastoma (GBM). In a genomic profiling of recurrent GBMs, the authors identify fusions involving the DNA repair enzyme MGMT, associated with a small percentage of tumors. They find that MGMT fusions are mutually exclusive with MGMT hypomethylation and global hypermutation, other established resistance mechanisms. They go on to perform functional analysis in U251 and U87 cells. They find that expression of MGMT fusions increases MGMT expression and decreases sensitivity to temozolomide, and that these phenomena are correlated. Moreover, MGMT fusion expression increases temozolomide resistance in a murine xenograft model. Finally, they demonstrate that MGMT fusions are shed in exosomes, and can be detected in peripheral blood, pointing to a potential method for clinical screening.

While the small number of MGMT-fused cases does detract somewhat from the significance of this work, it is a well-performed study describing a novel mechanism of resistance in a deadly tumor type.

My comments are below.

1) The reliance of the author's functional analyses on U87 and U251 cells is somewhat problematic. These lines have been subjected to extensive culture over the years and may not represent the best models of glioma biology. Many patient-derived glioma stem cells (GSCs) are currently available and recapitulate the broad spectrum of molecular features characterizing gliomas in general. The authors should recapitulate their key in vitro and in vivo findings using GSCs. The authors might also consider using the RCAS/tv-a system, which they have leveraged very effectively in the past, to address some of their hypotheses in vivo.

2) Recurrent glioma specimens are notoriously heterogeneous in their composition with regard to tumor content, reactive tissue, necrotizing treatment effect, and inflammatory cells. This heterogeneity could impact the detection of genomic factors associated with resistance, including MGMT-fusions. Can the authors provide more information as to the histopathological and molecular QC measures used in their profiling studies? Was there a tumor content threshold? Who did the analysis? What were the purity/ploidy measurements of their samples?

More minor

1) In the introduction, the authors should better explain the distinction between IDH-wild type and IDH-mutant glioma. General readers will not appreciate this difference. They should indicate early in the paper that malignant gliomas come in two basic varieties, IDH-mutant and IDH-wild type, and both are treated with temozolomide as the standard of care.

2) On a related note, the authors should consider presenting their genomic data more in the context of IDH-mutant vs IDH-wild type glioma, rather than HGG vs LGG.

Reviewer #2 (Remarks to the Author):

The manuscript "MGMT Genomic rearrangements..." submitted by Oldrini et al describes studies addressing a new form of MGMT genetic alteration in glioma that may contribute to TMZ resistance. In this study the authors show that in a small percentage of these patients, perhaps 3%, fusions of the MGMT gene with the 5' UTR of other genes results in higher level MGMT expression that in turn appears associated with TMZ resistance. In terms of novelty this is an

interesting and unique finding, and although as the authors note a similar MGMT fusion has been noted in a single medulloblastoma, the full range of such fusions in glioma has not been previously described. The work is also of interest in that there is growing interest in promoter highjacking and fusion transcripts as drivers in cancer, although this mechanism has not in the past been shown to involve MGMT. The work is also of importance as TMZ remains a mainline therapeutic for most forms of glioma, and resistance to TMZ and recurrence is common. As such the identification of a new mechanism of resistance and possibly a biomarker of this resistance, if even in a small percentage of patients is of interest to a broad range of investigators.

In terms of the studies themselves, they appear carefully done, appropriately controlled, and adequately statistically evaluated. There are no instances in which the data presented do not support the conclusions reached. I have only a few questions regarding the work as presented:

1. The authors note that they have analyzed "hypermethylation" in the samples and found that it was mutually exclusive with MGMT fusion. It is not clear however what type of "hypermethylation" the authors are considering. Is this hypermethylation the global type of hypermethylation noted in TMZ-treated recurrent GBM or are the authors talking about MGMT hypermethylation? If the latter, is this MGMT gene or promoter hypermethylation? One really can't tell from the text or from the description provided in the methods section. This is an important point as most readers would be interested in whether MGMT fusion was mutually exclusive with TMZ-induced genomic hypermethylation (as one might expect it to be).
2. I'm also a little unclear about the generation of the cells carrying MGMT fusions by CRISPR. For such experiments one would typically use CRISPR-Cas to create clones, then sort through the clones to find those with the desired recombination in one or both copies of the targeted genes. Line 122 in the text does not clearly describe how the process was carried out, and the suggestion that the end result of the initial engineering was "a mixed population of cells carrying different fusion events" is puzzling. Additionally the further use of populations of cells that arose as a result of TMZ selection is also a little puzzling as one would expect all the appropriately engineered cells to be TMZ resistant. This entire creation/selection process needs to be better described.

Reviewers' comments:

Reviewer #1 (Remarks to the Author):

This manuscript describes a potential resistance mechanism in temozolomide-treated glioblastoma (GBM). In a genomic profiling of recurrent GBMs, the authors identify fusions involving the DNA repair enzyme MGMT, associated with a small percentage of tumors. They find that MGMT fusions are mutually exclusive with MGMT hypomethylation and global hypermutation, other established resistance mechanisms. They go on to perform functional analysis in U251 and U87 cells. They find that expression of MGMT fusions increases MGMT expression and decreases sensitivity to temozolomide, and that these phenomena are correlated. Moreover, MGMT fusion expression increases temozolomide resistance in a murine xenograft model. Finally, they demonstrate that MGMT fusions are shed in exosomes, and can be detected in peripheral blood, pointing to a potential method for clinical screening.

While the small number of MGMT-fused cases does detract somewhat from the significance of this work, it is a well-performed study describing a novel mechanism of resistance in a deadly tumor type.

We thank the reviewer for acknowledging the quality and the novelty of our work. The MGMT-fused cases are certainly a relatively small number of glioma patients, approximately 3-4% of the whole cohort analyzed. However, the number goes up to 10-15% of the non-hypomethylated/ non-hypermutated patient population and thus we believe indeed it represents an important previously unidentified mechanism of TMZ resistance.

My comments are below.

1) The reliance of the author's functional analyses on U87 and U251 cells is somewhat problematic. These lines have been subjected to extensive culture over the years and may not represent the best models of glioma biology. Many patient-derived glioma stem cells (GSCs) are currently available and recapitulate the broad spectrum of molecular features characterizing gliomas in general. The authors should recapitulate their key in vitro and in vivo findings using GSCs. The authors might also consider using the

RCAS/tv-a system, which they have leveraged very effectively in the past, to address some of their hypotheses in vivo.

U87 and U251 cells are extensively used in the glioma field and their response to TMZ has been very well characterized by many different laboratories. However, we do agree with the reviewer, that they might not be the best model to recapitulate all the aspects of glioma biology and as requested we have repeated key experiments in patient-derived glioma stem cells. We have transduced the h543, a well characterized patient-derived tumor sphere line generated at the Memorial Sloan Kettering, with grRNA pairs for the BTRC-MGMT and the SAR1A-MGMT fusions. We confirmed the h543 cells carrying those fusions express high levels of MGMT (Figure S7d-e) and are resistant to TMZ *in vitro* (Figure S7f-g) and *in vivo* (Figure S7h). Considering the positive results obtained with the human h543 xenograft model we haven't attempted to generate further mouse models for the MGMT fusions using the much more challenging RCAS/tv-a system.

2) Recurrent glioma specimens are notoriously heterogeneous in their composition with regard to tumor content, reactive tissue, necrotizing treatment effect, and inflammatory cells. This heterogeneity could impact the detection of genomic factors associated with resistance, including MGMT-fusions. Can the authors provide more information as to the histopathological and molecular QC measures used in their profiling studies? Was there a tumor content threshold? Who did the analysis? What were the purity/ploidy measurements of their samples?

We thank the reviewer for this constructive suggestion. We agree that tumor heterogeneity might impact the detection of *MGMT* fusion. Low purity tumor may fail in detection of *MGMT* fusion, leading to underestimation of *MGMT* fusion frequency. In the collection of the CGGA samples, the pathologists excluded samples for which they felt to be of low purity, but this judgement was subjective and qualitative. We therefore performed an estimation of the tumor purity using a unified pipeline.

Many methods have been developed for tumor purity estimation, among which ABSOLUTE, FACETS, and ESTIMATE are widely used. Since ABSOLUTE and FACETS are developed for DNA sequencing data analysis, we run ESTIMATE to analyze our RNA sequencing data. The results showed that the tumor purity in our cohort ranged from 0.454 to 1, with median value as 0.861, and 250/252 (99.2%) samples have

purity >0.5(Figure R1a). We then compared the purity of MGMT-fusion-positive samples versus the MGMT-fusion-negative samples. We observed that MGMT-fusion-positive samples tends to be of higher purity, but the difference is not statistically significant (P=0.06, Wilcoxon's rank-sum test, Figure R1b).

Figure R1. CGGA samples purity

More minor

1) In the introduction, the authors should better explain the distinction between IDH-wild type and IDH-mutant glioma. General readers will not appreciate this difference. They should indicate early in the paper that malignant gliomas come in two basic varieties, IDH-mutant and IDH-wild type, and both are treated with temozolomide as the standard of care.

We thank the reviewer for this suggestion. We have now discussed in the introduction the distinction between the two types of gliomas (page 3, lines 53-57).

2) On a related note, the authors should consider presenting their genomic data more in the context of IDH-mutant vs IDH-wild type glioma, rather than HGG vs LGG.

As requested, in Panel 1a, and in the main text (page 4, lines 68-72), we have now removed the classification of the tumors as HGG and LGG and grouped by IDH mutation and 1p/19q co-deletion status.

Reviewer #2 (Remarks to the Author):

The manuscript “MGMT Genomic rearrangements...” submitted by Oldrini et al describes studies addressing a new form of MGMT genetic alteration in glioma that may contribute to TMZ resistance. In this study the authors show that in a small percentage of these patients, perhaps 3%, fusions of the MGMT gene with the 5’ UTR of other genes results in higher level MGMT expression that in turn appears associated with TMZ resistance. In terms of novelty this is an interesting and unique finding, and although as the authors note a similar MGMT fusion has been noted in a single medulloblastoma, the full range of such fusions in glioma has not been previously described. The work is also of interest in that there is growing interest in promoter hijacking and fusion transcripts as drivers in cancer, although this mechanism has not in the past been shown to involve MGMT. The work is also of importance as TMZ remains a mainline therapeutic for most forms of glioma, and resistance to TMZ and recurrence is common. As such the identification of a new mechanism of resistance and possibly a biomarker of this resistance, if even in a small percentage of patients is of interest to a broad range of investigators.

In terms of the studies themselves, they appear carefully done, appropriately controlled, and adequately statistically evaluated. There are no instances in which the data presented do not support the conclusions reached.

We are grateful to this reviewer for recognizing the broad interest of our findings and the quality of our work.

I have only a few questions regarding the work as presented:

1. The authors note that they have analyzed “hypermethylation” in the samples and found that it was mutually exclusive with MGMT fusion. It is not clear however what type of “hypermethylation” the authors are considering. Is this hypermethylation the global type of hypermethylation noted in TMZ-treated recurrent GBM or are the authors talking about MGMT hypermethylation? If the latter, is this MGMT gene or promoter hypermethylation? One really cant tell from the text or from the description provided in the methods section. This is an important point as most readers would be interested in whether MGMT fusion was mutually exclusive with TMZ-induced genomic hypermethylation (as one might expect it to be).

We apologize if it was not clear that “hypermutation” that was referring to TMZ-induced genomic hypermutation and not MGMT gene or promoter hypermutation. We have now specified it in the introduction.

2. I'm also a little unclear about the generation of the cells carrying MGMT fusions by CRISPR. For such experiments one would typically use CRISPR-Cas to create clones, then sort through the clones to find those with the desired recombination in one or both copies of the targeted genes. Line 122 in the text does not clearly describe how the process was carried out, and the suggestion that the end result of the initial engineering was “a mixed population of cells carrying different fusion events” is puzzling. Additionally the further use of populations of cells that arose as a result of TMZ selection is also a little puzzling as one would expect all the appropriately engineered cells to be TMZ resistant. This entire creation/selection process needs to be better described.

We apologize for not having described in depth the generation of the MGMT fusion cell lines resistant to TMZ. We have now edited the main text to make it clearer (page 7, lines 129-137). We have also included the details in the method section (page 20, lines 478-481). As compared to classical CRISPR/Cas9 editing by a single gRNA, generating a gene fusion/translocation by expressing gRNA pairs is a much less efficient process. Approximately 5-10% of the cells might carry the desired edits, as compared to 50-90% of classical CRISPR/Cas9. We reasoned that we could indeed increase the number of positive clones carrying the desired gene fusion by selecting the cells with TMZ, similarly to what could happen in the TMZ-treated patients. Although, there might be the concern that we might select also for other TMZ-resistance mechanisms, with our experimental settings we never observed resistant clones in the control gRNA transduced cells, suggesting that we could successfully isolate the clones expressing the *MGMT* fusion of interest.

REVIEWERS' COMMENTS:

Reviewer #1 (Remarks to the Author):

I have no further issues with this manuscript. I commend the authors on a job well done.

Reviewer #2 (Remarks to the Author):

my previous concerns have been adequately addressed